# Comment on Redmayne, M.; Maisch, D.R. ICNIRP Guidelines’ Exposure Assessment Method for 5G Millimetre Wave Radiation May Trigger Adverse Effects. *Int. J. Environ. Res. Public Health* 2023, *20*, 5267

**DOI:** 10.3390/ijerph20227029

**Published:** 2023-11-07

**Authors:** Kenneth R. Foster, Quirino Balzano

**Affiliations:** 1Department of Bioengineering, University of Pennsylvania, Philadelphia, PA 19106, USA; 2Independent Researcher, Annapolis, MD 21401, USA; quibalzano@gmail.com

**Keywords:** Brillouin precursors, 5G new radio, thermal hazards

## Abstract

This article discusses the contention in the commented-upon paper that Brillouin precursors generated by 5G New Radio (5G NR) and other cellular systems are a possible cause of tissue damage at deeper layers of tissue than the power penetration depth of the carrier frequency. The original theory for Brillouin precursors from pulsed radiofrequency signals (RF-EMF) and speculation about their possible health effects dates back to the 1990’s and was based on studies of the propagation of very short (nanosecond) ultrawide-bandwidth RF pulses through water. This assumption is not correct for cellular telephone signals due to their narrow bandwidth. The commented-on paper provides no alternative rationale as to why Brillouin effects should cause tissue damage from RF-EMF radiation from cellular and other communications systems. Other inaccuracies in this paper concerning thermal responses of tissue to RF-EMF are also noted.

## 1. Introduction

Redmayne and Maisch [1] challenge the safety of current International Commission on Nonionizing Radiation Protection (ICNIRP) guidelines for exposure to pulsed radiofrequency (RF-EMF) exposure from 5G NR and other communications system “in part due to possible Brillouin precursor pulse formation”. They suggest that the [presence of Brillouin precursors] “changes the expected extent and depth of exposure; it does not increase the total amount of energy, but maintains some of it as kinetic energy, redistributes the frequency spectrum of the original signal and carries the energy deeper into the dispersive living tissue of the body than predicted by conventional thermal models”.

Apart from the lack of a clear rationale as to why this should constitute a hazard, simple considerations based on the bandwidth of cellular telephone signals show that the theoretical model that the authors assume for Brillouin precursors is simply inapplicable to 5G NR or other cellular telephone signals. Other deficiencies are noted as well.

## 2. Brillouin Precursors

### 2.1. Origin of Phenomenon

Brillouin precursors arise when a broadband electromagnetic wave packet propagates through a dispersive medium. The medium acts as a lowpass filter, removing higher frequency components from the wave packet as it passes through it. Lower frequency components, which are attenuated less strongly than the carrier, will propagate more deeply into the medium. As it progresses through the medium, the packet changes shape, typically showing peaks at the beginning and end of the packet (Brillouin precursors). Brillouin precursors have been examined theoretically by Oughstun and others (e.g., [2]), and have been experimentally observed by Stoudt et al. [3], Dawood et al. [4] and others, confirming the theory. 

Redmayne and Maisch refer to work by Albanese et al. [5], who examined the propagation of “ultrashort” (1 ns) pulses of RF-EMF with a carrier frequency of 10 GHz in water. Those authors proposed that Brillouin precursors could “create a disturbance in tissue at greater depth than would be expected if one were to use the depth of penetration of the basic carrier frequency” [5]. This claim forms the basis of Redmayne and Maisch’s critique. The “greater depth” refers to the low-frequency components of the wave packet, which represent only a small fraction of the total energy carried by the pulse. 

Albanese recommended “zero human exposure to such unique precursor and gendering pulses” [6]. Albanese’s claims were challenged by Adair [7,8] and Stoudt et al. [3], but nevertheless have been repeated on activists’ websites. Now, Redmayne and Maisch have resurrected the theory even though they do not explain how such precursors might cause tissue damage.

Simple considerations show that Albanese’s original model is irrelevant to band-limited communications signals such as from cellular communications systems. We consider a pulse of RF-EMF with the same parameters as in Albanese et al. [6], consisting of 10 cycles of 10 GHz radiation that is normally incident on the medium (water); all of the incident energy is assumed to be transmitted into the medium. This pulse is represented by a carrier wave of frequency *ω*_c_ = 2π10^10^ rad/sec and amplitude E_c_ (measured just beneath the surface of the medium) that is modulated by a square wave of duration t_p_ (1 ns). The surface of the medium is located at z = 0, and increasing z is directed into the medium. 

The wave packet immediately below the surface can be written as
(1)$$E(t,0)=E_{c}r\mathrm{ect}(t/t_{p})\cos(\omega_{c}t)$$
where rect(t/t_p_) is a rectangular function that is equal to 1 for |t| < t_p_/2, ½ for |t| = t_p_/2, and 0 otherwise. This “ultrashort” pulse is vastly different, both in duration and in frequency spectrum, from 5G or other communications signals but may be a reasonable approximation for other problems (e.g., ultrawide-band radar or brief laser light pulses). 

### 2.2. Frequency Spectrum of Pulse

The Fourier transform of Equation (1) is
(2)$$\begin{array}{ll} E\left(\omega ,0\right) & =2E_{c}\omega_{c}\frac{\sin\omega N\frac{T_{c}}{2}}{\omega^{2}-\omega_{c}^{2}}\\ & =2E_{c}\omega_{c}\frac{\sin(\omega \frac{t_{p}}{2})}{\omega^{2}-\omega_{c}^{2}} \end{array}$$
where T_c_ is the period of the wave.

The magnitude spectrum |E(ω,0)| at the surface consists of a main lobe at 10 GHz, with an infinite number of sidelobes spaced 1 GHz apart that progressively decrease in magnitude with increasing distance from the main lobe. 

The attenuation of different frequency components of the pulse can be analyzed in terms of the frequency components of the wave packet. We consider a plane electromagnetic wave of radian frequency *ω* as it propagates through an absorbing medium. The electric field E (*ω*,z) of the wave at depth z is related to that at the surface E (*ω*,0) via
(3)$$E(\omega ,z)=E(\omega ,0)e^{-(\alpha +j\beta )z}e^{j\omega t}$$
where
(4)$$\alpha +j\beta =\frac{j\omega \sqrt{\varepsilon^{*}}}{c}$$
and α and β are, respectively, the attenuation and phase constants of the medium, c is the velocity of light in vacuum, and ε* is the complex permittivity of water. We consider related quantities instead: the phase velocity v = ω/β and the power penetration depth L = 1/(2α) (the depth at which the power density of the wave has decreased by a factor 1/e compared to that at the surface). 

For water at room temperature at frequencies below ≈100 GHz, ε* is closely described using the Debye function
(5)$$\varepsilon^{*}=\varepsilon_{\infty }+\frac{\varepsilon_{s}-\varepsilon_{\infty }}{1+j\omega /\omega_{c}}$$
where ε_s_ = 78.6, ε_∞_ = 5.4, and ω_c_ = 2π●(19 GHz) rad/s at room temperature. The wave velocity and power attenuation depth for water are shown in Figure 1.

Figure 2 shows the magnitude spectrum of the 1 ns “ultrashort” pulse as described by Albanese et al. [5,6] (10 cycles at 10 GHz) with rectangular modulation given in Equation (1). The main lobe is at 10 GHz, corresponding to the carrier frequency, and the lowest frequency sidelobe is close to 0.5 GHz. For water, the power penetration depth (L) ranges from 44 cm for the lowest frequency sidelobe to 1.3 mm at the carrier frequency, decreasing further at higher frequencies. 

Thus, as the pulse propagates through the medium, higher frequency components will be increasingly filtered out of the wave packet, and the evolution of the pulse will increasingly be determined by the attenuation characteristics of its lowest frequency components. The “deeper penetration depths” referred to by Redmayne and Maisch, and previously by Albanese, refer to the comparatively deeper penetration of the lowest frequency components of the wave packet, which represent only a minor fraction of the total energy in the packet. Cellular and other communications waveforms are strictly band-limited and consequently lack such “deeply penetrating” components. All frequency components in a 5G NR signal are contained within the channel bandwidth and consequently their power penetration depths are similar to that of the carrier. Moreover, in the modulation scheme used by 5G NR (orthogonal frequency-division multiplexing or OFDM), signals are transmitted over many subcarriers, each having a very narrow bandwidth (15 kHz), that are distributed over the channel bandwidth. 

Redmayne and Hirsch state that Brillouin precursors “cover [] the full spectrum of frequencies from −∞ to ∞”. That is only true for the idealized waveform (Equation (1)), which has discontinuities in the slope of the modulating function at each end of the pulse. They state “the likelihood of producing Brillouin precursors increases with extra transmission speed, increasing the pulse rate into Gbps (billion bits per second), and with a GHz bandwidth of more than 500 MHz”. In fact, 5G NR (as well as other communications signals) are strictly band-limited, both by the spectrum available to the carrier and (for cellular communications) by the slice of spectrum (the channel) that the carrier has assigned to the user. 

For the case of 5G NR, higher data transmission speeds are accomplished through use of wider channels as provided by the 5G NR standard, taking care to ensure that the signal remains within the frequency band assigned to the carrier. Moreover, waveforms of 5G NR and other digitally modulated signals are more complicated than a series of square-wave-modulated pulses as in Equation (1). To suppress intersymbol interference and other undesirable effects from sidebands, designers of 5G equipment include sophisticated filtering algorithms [9]. In addition, a signal radiated from any personal cellular communication device or other transmitter is limited in bandwidth by the electronic circuits generating the RF signals, the transmission lines feeding the antennas, and the antennas themselves. Thus, the square-wave-modulated RF-EMF pulse in Equation (1) with infinite bandwidth is simply irrelevant to cellular communications signals.

### 2.3. Bandwidth of 5G NR Signals

Table 1 summarizes several 5G NR (5G New Radio) bands allocated (in the U.S.) by the Federal Communications Commissions, and the bandwidths of channels in each band defined by the 5G NR standard. Also shown are the power penetration depths and variation in L across the widest channel in each band. The instantaneous bandwidth of the channels (which is at most the channel width) is always a small fraction of the carrier frequency, which constrains the rise time of any pulsed signals to nonzero values. At the frequencies used by 5G NR, the power penetration depth at the center of a band L is small, and its variation ΔL across each channel/band is small relative to L. The narrow bandwidth of 5G NR limits, which are due to several constraints and must be rigorously maintained to allow for the system to work properly, prevents the formation of Brillouin precursors from 5G NR devices. 

## 3. Conclusions

Albanese’s calculations, which pertain to the ultrashort pulses of a nominally infinite bandwidth, are simply irrelevant to wireless communications, which are strictly band-limited, and particularly to 5G NR with its use of OFDM modulation. If Redmayne and Maisch have something else in mind, they should present a quantitative model and explain in detail just how they envision tissue damage being produced. 

## 4. Other Comment

An extensive body of literature now exists on human exposure to 5G signals using both numerical modeling and onsite measurements [11,12,13,14] as well as modeling studies of thermal responses of tissue to RF-EMF exposure at 5G NR high-band and MMW frequencies. Redmayne and Maisch provide a broad critique of ICNIRP guidelines based on a selective reading of selected references. 

However, their critique is superficial and misleading in several respects. The authors included a quotation from [15] on the possibility of “intense surface heating and a steep, rapid rise in temperature” from pulsed MMWs but did not note that the original authors had made an orders-of-magnitude error in estimating the near-surface heating of tissue from such exposures [16]. This speculation is irrelevant to 5G NR in any event.

As another example, Redmayne and Maisch assert that skin modeling studies “treat the skin as homogeneous dermal tissue” to justify what they consider limitations in the ICNIRP exposure guidelines. That is not true at all. Thermal modeling studies have used models ranging from simple homogeneous tissue models, to multilayer models of skin and subcutaneous tissues [17], to high-resolution-image-based models of the body [18]. For an extensive review of this literature, see [19]. 

In their review, Funahashi et al. [20] concluded that “…TPD [transmitted power density into skin, the dosimetric quantity used in ICNIRP local exposure guidelines above 6 GHz] provides an excellent estimate of skin temperature elevation through the millimeter-wave band (30–300 GHz) and a reasonable and conservative estimate down to 10 GHz, whereas the SAR [specific absorption rate, in W/kg] is a good metric below 3 GHz”. 

Many open research issues remain, for example those related to the experimental (as opposed to numerical modeling) assessment of absorbed power in the body from near-field exposures above 6 GHz, which are intended to improve compliance assessment and may possibly lead to refinement of the ICNIRP guidelines for local-body exposures themselves [21]. 

A more informed and complete review than Redmayne and Maisch provide is needed to help readers understand the current state of knowledge and unresolved questions about these important but complicated issues.

## Figures and Tables

**Figure 1 ijerph-20-07029-f001:**
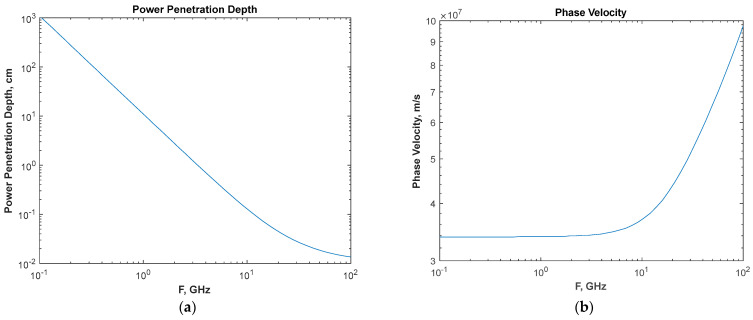
Power penetration depth (**a**) and phase velocity (**b**) of a plane wave propagating through pure water.

**Figure 2 ijerph-20-07029-f002:**
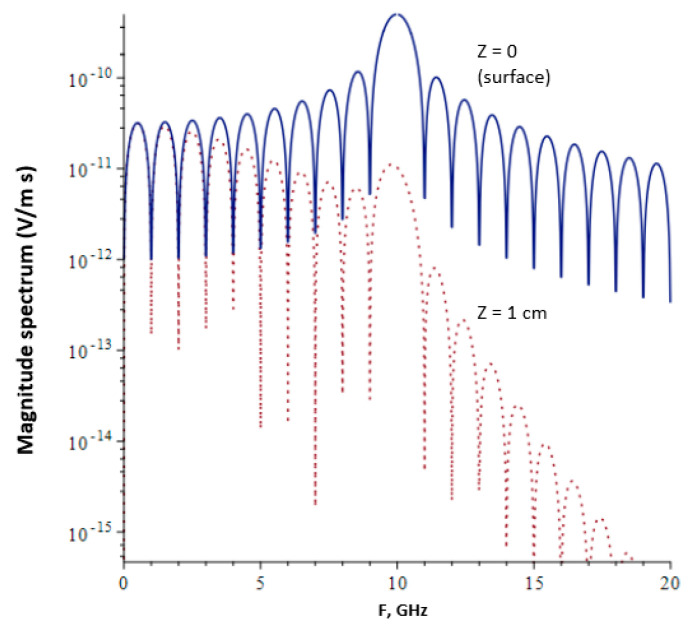
Frequency spectrum (magnitude) of a 1 ns, square-wave-modulated RF-EMF pulse with carrier frequency of 10 GHz propagating in pure water at z = 0 (surface) () and z = 1 cm below the surface (^…^). The wave is normally incident on the surface, and 100% transmission into the medium is assumed. The power penetration depth of a frequency component at 3 GHz in pure water is about 1 cm, meaning that higher frequency components are significantly attenuated even at a depth of 1 cm beneath the surface.

**Table 1 ijerph-20-07029-t001:** Frequency bands and channel bandwidths used by 5G NR systems.

5G Band	Uplink Frequencies (GHz)	Downlink Frequencies (GHz)	Channel Bandwidths (MHz) *	Power Penetration Depth L in Skin/Muscle, cm (at the Center of the Band), cm	ΔL in Widest Band
N5	0.824–0.849	0.869–0.894	5–25	2.1 (muscle)	9 mm
N66	1.710–1.780	2.100–2.200	5–45	1.5 (muscle)	0.3 mm
N2	1.850–1.910	1.930–1.990	5–40	1.4 (muscle)	0.2 mm
n258	24.25–27.50		50–400	0.05 (skin)	0.01 mm

* Uplink: from handset to base station; downlink from base station to handset. Channel bandwidth varies with implementation. Frequency bands as used in the U.S.; a more extensive tabulation can be found at https://en.wikipedia.org/wiki/5G_NR_frequency_bands (accessed on 18 May 2022). The power penetration depth L is the distance at which power density is reduced by a factor of 1/e below that at the surface. For tissue, these values were calculated from data by Gabriel et al. [10].

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
