# Peer review of "Comment on Redmayne, M.; Maisch, D.R. ICNIRP Guidelines’ Exposure Assessment Method for 5G Millimetre Wave Radiation May Trigger Adverse Effects. Int. J. Environ. Res. Public Health 2023, 20, 5267"

_ijerph, 2023, doi:10.3390/ijerph20227029_

Round 1
Reviewer 1 Report
Comments and Suggestions for Authors
The manuscript consists of total 7 pages, including 2 figures, 1 table and total 11 literature references. The polemical manuscript concerns potential biological adverse effects that according to the previous article of other Authors is supposed to result from 5G NR radiation exposure. As such, it is current and, while the topic rises many discussions, it is likely to be interesting to the Journal's Readers and it is within the scope of works published in the Journal. The English language quality of the text is acceptable.
The Authors refer clearly to the challenged fragments of the other Authors articles, in a merit and balanced way. Their line of argumentation is clear enough and easy to follow.
I feel that, for the sake of keeping the right structure organization, the currently very brief introduction shall be enriched with the fragments of the text describing Brillouin precursors general theory and nature (the fragments of the text may be shifted into the Introduction section), while the following section may contain the parts directly associated with the polemic.
The Conclusion section wraps-up the postulates of the polemic in a concise statement.
The "other comment" section adds additional information relevant to the background of the polemic's topic.
The argumentation is enriched with figures and a table, reasonably aiding to the line of argumentation.
The standard of literature references differs from the MDPI accepted notation. The literature references are reasonably numerous and related to the topic of the text. The Authors may consider enriching the introductory statements by adding information on not only theoretical but also field-detected radiation exposures and responses as in e.c. https://doi.org/10.3390/environments7030022 , https://doi.org/10.3390/app11083592, https://doi.org/10.3390/app10155280 , https://doi.org/10.3390/app10175971 , https://doi.org/10.3390/electronics9020223 .
There is an excessive double "and" in lines 132-133 (before and after the sentence in brackets).
In the line 222 there are typing errors in the beginning of literature reference.
Author Response
the requested revisions were made:
1, The additional references are added, in the concluding section together with a few other related references
2. Minor corrections and edits to the endnotes (which were formatted in Endnote using the template chicago-mdpi
3. A couple additional sentences added to the beginning quoting more of Redmayne and Maisch
also, some minor text polishing throughout